# Randomised controlled feasibility trial of real versus sham repetitive transcranial magnetic stimulation treatment in adults with severe and enduring anorexia nervosa: the TIARA study

Bethan Dalton,[1] Savani Bartholdy,[1] Jessica McClelland,[1] Maria Kekic,[1] Samantha J Rennalls,[2] Jessica Werthmann,[1] Ben Carter,[3] Owen G O'Daly,[2] Iain C Campbell,[1] Anthony S David,[4] Danielle Glennon,[5] Nikola Kern,[5] Ulrike Schmidt[1,5]

BD, SB and JM contributed equally.

For numbered affiliations see end of article.

**Correspondence to**
Professor Ulrike Schmidt;
ulrike.schmidt@kcl.ac.uk

## ABSTRACT

**Objective** Treatment options for severe, enduring anorexia nervosa (SE-AN) are limited. Non-invasive neuromodulation is a promising emerging intervention. Our study is a feasibility randomised controlled trial of repetitive transcranial magnetic stimulation (rTMS) in individuals with SE-AN, which aims to inform the design of a future large-scale trial.

**Design** Double-blind, parallel group, two-arm, sham-controlled trial.

**Setting** Specialist eating disorders centre.

**Participants** Community-dwelling people with anorexia nervosa, an illness duration of ≥3 years and at least one previous completed treatment.

**Interventions** Participants received 20 sessions (administered over 4 weeks) of MRI-guided real or sham high-frequency rTMS to the left dorsolateral prefrontal cortex in addition to treatment-as-usual.

**Outcomes** Primary outcomes were recruitment, attendance and retention rates. Secondary outcomes included body mass index (BMI), eating disorder symptoms, mood, quality of life and rTMS safety and tolerability. Assessments were conducted at baseline, post-treatment and follow-up (ie, at 0 month, 1 month and 4 months post-randomisation).

**Results** Thirty-four participants (17 per group) were randomly allocated to real or sham rTMS. One participant per group was withdrawn prior to the intervention due to safety concerns. Two participants (both receiving sham) did not complete the treatment. rTMS was safe and well tolerated. Between-group effect sizes of change scores (baseline to follow-up) were small for BMI ($d$=0.2, 95% CI −0.49 to 0.90) and eating disorder symptoms ($d$=0.1, 95% CI −0.60 to 0.79), medium for quality of life and moderate to large ($d$=0.61 to 1.0) for mood outcomes, all favouring rTMS over sham.

**Conclusions** The treatment protocol is feasible and acceptable to participants. Outcomes provide preliminary evidence for the therapeutic potential of rTMS in SE-AN.

### Strengths and limitations of this study

► This is the first randomised controlled feasibility trial of multi-session repetitive transcranial magnetic stimulation (rTMS) treatment in people with anorexia nervosa (AN).

► It targeted those with severe and enduring AN (SE-AN).

► A range of outcomes were assessed (recruitment, retention, safety, tolerability and effect sizes of clinical outcome variables); thus, it provides useful data for implementing a larger scale randomised controlled trial of rTMS in SE-AN.

► The study had a small sample size, was not powered or designed to assess the efficacy of rTMS in SE-AN and the follow-up duration was short.

Largest effects were observed on variables assessing mood. This study supports the need for a larger confirmatory trial to evaluate the effectiveness of multi-session rTMS in SE-AN. Future studies should include a longer follow-up period and an assessment of cost-effectiveness.

**Trial registration number** ISRCTN14329415; Pre-results.

## INTRODUCTION

Anorexia nervosa (AN) is a life-threatening disorder of multifactorial aetiology. Alterations in neural circuits involved in reward processing, negative affect and stress, appetite regulation, cognitive (self-regulatory) control and socioemotional processes have been implicated in its causation and maintenance.[1 2]

Approximately 20% of patients develop a severe, enduring form of illness (SE-AN).[3]

These patients typically have high levels of depression and anxiety, are socially isolated and markedly impaired in their ability to care for themselves. In fact, their quality of life is comparable with that of patients with depression and impairments in social contact and self-care are comparable with those in psychosis.[4] Recent research on treatments for AN has mainly focused on psychological therapies.[5] Only two small trials have focused on SE-AN, using psychological therapies or medication to improve quality of life[6] or achieve weight gain, although with limited success.[7] Thus, there is a need for novel interventions for this group.[1 5 6 8]

Non-invasive neuromodulation is a promising emerging treatment for SE-AN,[2 5 9] in particular repetitive transcranial magnetic stimulation (rTMS) (eg, [10 11]). rTMS can enhance (high-frequency) or inhibit (low-frequency) cortical activity in targeted brain areas. It appears to increase neuroplasticity, and hence may be of value in chronic or treatment-resistant neurocircuit-based disorders, such as SE-AN.[9] Based on the Research Domain Criteria, candidate targets for rTMS in eating disorders (EDs) have been described, involving brain structures/circuitry in the cognitive control, positive and negative valence, and social processes systems.[12] Partly for theoretical reasons, but also for accessibility reasons, rTMS studies have targeted the dorsolateral prefrontal cortex (DLPFC) or the dorsomedial prefrontal cortex.[12]

Proof-of-concept studies have shown that rTMS is a promising treatment in AN.[9 12] We previously carried out two single-session studies in AN[13 14] and a case series of 20 sessions of rTMS in SE-AN,[11 15] all involving high-frequency rTMS to the left DLPFC. These studies showed that rTMS can lead to both short-term and long-term improvements in ED symptoms, mood and reward-based decision making. Thus, there is a rationale for further exploring the therapeutic potential of rTMS in SE-AN.

To date, no sham-controlled randomised controlled trial (RCT) of rTMS in SE-AN has been conducted. The present trial (Transcranial Magnetic Stimulation and Neuroimaging in Anorexia Nervosa) aimed to assess the feasibility of using rTMS compared with sham treatment in people with SE-AN and to inform the development of a large-scale sham-controlled RCT.[16] Our primary objective was:

a. To assess recruitment, attendance and retention rates. Secondary objectives were:

a. To estimate the treatment effect sizes and standard deviations for outcome measures to inform future sample size calculations.

b. To determine safety and tolerability of rTMS in SE-AN.

Subsidiary objectives were to assess neuropsychological and neural correlates and predictors of rTMS treatment in AN and to assess within-session change processes. Findings relating to these will be published elsewhere. The study rationale, aims and tentative hypotheses, along with the trial design and methodology are described fully in a protocol paper.[16]

## METHODS
### Design, participants and setting

In a double-blind parallel group, randomised controlled design, participants were allocated to receive 20 sessions of either real or sham high-frequency rTMS in addition to treatment-as-usual (TAU). Outcomes were assessed at baseline (pre-randomisation), post-treatment (~1 month post-randomisation) and at follow-up (~4 months post-randomisation).

Right-handed community-dwelling adults (≥18 years old) with a current Diagnostic and Statistical Manual of Mental Disorders (5th edition[17] [DSM-5]) diagnosis of AN and a body mass index (BMI) >14 kg/m$^2$ were eligible. Participants had to have a severe, enduring form of AN; this was defined as an illness duration of ≥3 years and completion of at least one previous course of treatment (eg, National Institute for Health and Care Excellence [NICE[18]]-recommended specialist psychotherapy or specialist day-patient or inpatient treatment for their ED) (We accept that there is a continuing debate on definitions of SE-AN; for review, see Broomfield *et al.*[19]). To take part, participants needed agreement from their ED clinician or general practitioner. Main exclusion criteria were related to contraindications to either rTMS or MRI (for details, see [16]).

Participants were recruited from the Eating Disorders Unit at the South London and Maudsley NHS Foundation Trust, through online and media advertisements and through participation in other research projects.

Potential participants underwent a screening procedure to determine eligibility (see [16] for details). Once eligibility was determined, participants' written informed consent was obtained prior to the baseline assessment.

### Randomisation and blinding

Randomisation was conducted by the King's College London (KCL) Clinical Trials Unit (CTU) using their automatic online system. Randomisation requests were submitted by study researchers via the web-based CTU system after the baseline assessment. Participants were allocated at a ratio of 1:1 to the two trial arms using a restricted stratified randomisation algorithm stratifying by prognostic factors: AN subtype (AN restrictive or AN binge–purge) and intensity of TAU (high: day-patient treatment, or low: outpatient or no treatment). The stratification was implemented by minimised randomisation with a random component. The first n cases (n was not disclosed) were allocated entirely at random to further enhance allocation concealment.

Participants and researchers were blinded to treatment allocation, except for one researcher who conducted follow-up assessments and unblinded participants. For practical reasons, a small proportion of rTMS sessions (116/594 sessions; 19.53%) was delivered by the unblinded researcher. All other rTMS therapists remained blinded until study data had been collected and analysed. Participants were unblinded at 4 months post-randomisation once they had completed the study. Participants who

received the sham intervention were offered real rTMS (if they continued to meet eligibility criteria) after their follow-up. Assessments of blinding success were carried out for rTMS therapists and participants. For details, see online supplementary information.

## Intervention

Participants received 20 sessions of (real or sham) high-frequency rTMS to the left DLPFC over 20 consecutive weekdays, in addition to TAU (ie, specialist ED outpatient or day-patient treatment, or no current treatment). Each session lasted 30–60 minutes, including preparation time, 20 minutes of rTMS and administration of within-session measures. rTMS sessions were conducted at the Institute of Psychiatry, Psychology & Neuroscience, KCL, in a designated rTMS suite.

Prior to starting treatment, all participants underwent a structural MRI scan to localise the DLPFC (Talairach co-ordinates $x=-45, y=45, z=30$)[11 20] for the purpose of neuronavigation (using Brainsight neuronavigation software). To determine the intensity of the rTMS stimulation, a Magstim Rapid device (Magstim, Whitland, Wales, UK) with a real TMS figure-of-eight coil was used to determine participants' motor threshold (MT), which represents membrane-related excitability of cortical axons. Using the motor-evoked potential method, the MT was established by determining the minimum stimulator output intensity required to obtain 5 out of 10 motor-evoked potentials $>50 \mu V$. MT was acquired weekly for each participant to ensure accuracy of the rTMS dose.

The Magstim Rapid device and Magstim D70-mm air-cooled real and sham coils were used to administer real and sham rTMS. Participants in the real group received 20 sessions of high-frequency (10 Hz) rTMS at 110% of their individual MT, consisting of 20 5s trains with 55s inter-train intervals delivered to the left DLPFC (a total of 1000 pulses delivered over each 20 minute session).[11 20] Sham stimulation was administered at the same parameters as real rTMS; however, a sham coil was used. The sham coil produces the same noises and feelings as the real coil but does not deliver active stimulation to the brain, rather it stimulates facial and scalp nerves.

## Outcomes

The primary outcomes to assess feasibility were recruitment, attendance and retention rates. To judge whether or how to proceed with a future definitive trial, we prespecified two criteria: first, recruitment as planned (see protocol paper[16] and the 'Changes to planned protocol' section below), and second, research follow-up rates of ≥80% at 4 months post-randomisation. We did not prespecify any rTMS session attendance rates required for progression to a full trial, but clearly these would also guide a decision about the feasibility of a future trial. rTMS session attendance was recorded using a specially designed case record form.

Secondary feasibility outcomes included a range of clinical measures administered at baseline, 1 month (post-treatment) and 4 months post-randomisation (follow-up) to assess ED symptomatology, mood, other psychopathology and quality of life. Neurocognitive and neuroimaging assessments of rTMS treatment (see protocol paper[16]) were also completed but will be presented elsewhere.

### ED symptomatology

The outcome measures used were BMI, the Eating Disorder Examination Questionnaire version 6.0,[21] the Fear of Food Measure,[22] the Self-Starvation Scale[23] and the Eating Disorder Recovery Self-Efficacy Questionnaire.[24]

### Mood and other psychopathology

The measures used included the Depression, Anxiety and Stress Scale – 21 item (DASS-21),[25] the Positive and Negative Affect Schedule,[26] the Profile of Mood States[27] and the revised Obsessive-Compulsive Inventory.[28]

### Quality of life

The measures used were the EuroQol Quality of Life Scale (5-level EQ-5D version; EQ-5D-5L)[29] and the Clinical Impairment Assessment.[30 31]

In light of the prominent mood and quality of life component of SE-AN, and the association between these two variables in SE-AN,[4] the clinical outcome to be assessed as a primary outcome in a future definitive trial would most likely be the DASS.[25]

### Additional service utilisation

Patients' additional service utilisation was assessed with a self-report version of the Clinical Service Receipt Inventory[32] and a specially designed case record form.

### Safety, tolerability and participants' experience of treatment

To ensure safety, participants' weight, blood pressure (sitting and standing) and pulse were monitored weekly. Routine blood tests (including full blood count, urea and electrolytes and renal and liver function tests) were conducted prior to the start of rTMS treatment and were repeated at the midpoint of treatment or more frequently if clinically indicated. rTMS-associated side effects and participants' expectations and experience of treatment were also assessed (see online supplementary files).

### Procedure

Full details of the procedures and a table of measures-by-assessment are presented in our protocol paper.[16] All procedures were identical between groups, except for the rTMS intervention.

### Baseline assessment and rTMS sessions

Participants' weight and height were measured, and they completed a battery of questionnaires (described above) and neuropsychological computer tasks (not presented here). A 1-hour MRI scan was also conducted. This included a structural MRI (for rTMS target localisation), functional MRI (fMRI), resting state fMRI and arterial spin labelling (not reported here). Thereafter,

participants were randomly allocated to real or sham rTMS treatment.

All rTMS procedures and parameters were in accordance with the current safety and application guidelines for rTMS.[33] Treatment was delivered by researchers trained in rTMS administration.

Each rTMS session (except session 1) started with assessment of any side effects experienced since the previous session. Within-session ED cognitions were measured with VAS (relating to subsidiary aims, published separately), completed following brief cue exposure (ie, film clip of highly palatable foods) immediately before and after each rTMS session.

### Post-treatment assessment (1 month post-randomisation)

The post-treatment assessment occurred within 1 week of the final rTMS session and included the same elements as the baseline assessment.

### Follow-up (4 months post-randomisation)

This final assessment repeated the post-treatment assessment, except no MRI scan was conducted. In this session, an audio-recorded qualitative semi-structured interview was undertaken to ascertain participants' views on and experience of rTMS (published in full elsewhere), and blinding success was evaluated. Participants were then unblinded and individuals in the sham rTMS group were offered real rTMS treatment.

### Changes to planned protocol

We planned to recruit 44 participants but revised this to 30 participants because a greater than anticipated proportion of potential participants were not eligible (eg, due to MRI/rTMS contraindications or being left-handed). These figures are in line with recommendations for feasibility trials[34] and accounted for attrition. Additionally, we removed the upper BMI limit ($18.5 \, kg/m^2$) to reflect the change in diagnostic criteria for AN in DSM-5.[17]

### Patient and public involvement (PPI)

In preparation for the present study, we asked participants in our previous proof-of-concept studies on rTMS in EDs whether they would be interested in undergoing a full course of rTMS treatment, with the vast majority (41/47; 87%) answering affirmatively.[14 35] In 2013, our early studies in rTMS were featured in a BBC TV documentary, and following this, approximately 50 people with AN or their relatives contacted us as they were keen to have rTMS treatment, even if they had to travel. Many people who contacted us had SE-AN, with multiple unsuccessful previous treatments. This shows that there is an unmet need in relation to treatments for SE-AN and that patients with SE-AN and their carers see rTMS as a treatment to be prioritised in research.

In planning the present study, discussions with patients/carers influenced our study design as follows: first, we were originally concerned that daily rTMS treatment over 4 weeks might be too burdensome. However, our PPI advisors thought this to be acceptable. Second, participant feedback emphasised the importance of including a broad range of outcome measures, rather than a narrow focus on weight and eating. Third, it encouraged us to include a sham control condition in the study so as to not create unfounded expectations of success that may be based on a placebo response. The completed study protocol was reviewed and enthusiastically endorsed by one person with AN who had participated in our previous rTMS case series[11] and another made minor comments, which were incorporated.

Participant experience of rTMS treatment and other aspects of the current study, including assessment and treatment burden, were assessed with qualitative interviews at follow-up, as briefly described above. Data will be reported elsewhere and will inform future rTMS trials in AN.

An expert by experience and a carer of a person with AN were part of our trial steering group and reviewed and advised on the conduct of the study, its dissemination and future trial design. A summary of the results of the study has been sent to all study participants, and they will be provided with a copy of this article.

### Data analysis

Primary feasibility outcomes are presented as n/N (%). The post-treatment and follow-up group means and standard deviations (SD) for secondary outcomes were adjusted for baseline and presented with effect sizes (Cohen's $d$) alongside 95% confidence intervals (CIs). Last observation carried forward imputation was used for missing data.

## RESULTS

### Patient flow, attendance and retention

Patient flow is shown in the Consolidated Standards of Reporting Trials diagram (figure 1), and the primary feasibility outcome findings are described below. The trial duration was determined by the funding period.

During the 20-month recruitment period (August 2015–March 2017), 269 people expressed interest in the study. Of these, 61 (22.7%) did not meet the inclusion criteria and 81 (30.1%) were not interested and/or declined to participate, with the majority citing trial practicalities as a reason for this (eg, accessibility, financial limitations, time commitment). Thirty-four people were enrolled and randomly allocated to the two treatment arms (n=17 per group). Two randomised participants were withdrawn for safety reasons prior to starting treatment: one participant (allocated to sham rTMS) had a syncope during her initial MT assessment; the other patient's (allocated to real rTMS) weight had dropped below BMI $14 \, kg/m^2$. These participants were excluded from the analyses. All others were included.

Thirty-two participants started treatment; two participants allocated to sham rTMS stopped treatment, one after four sessions (due to anxiety with travel) and one after nine sessions (due to multiple commitments). All

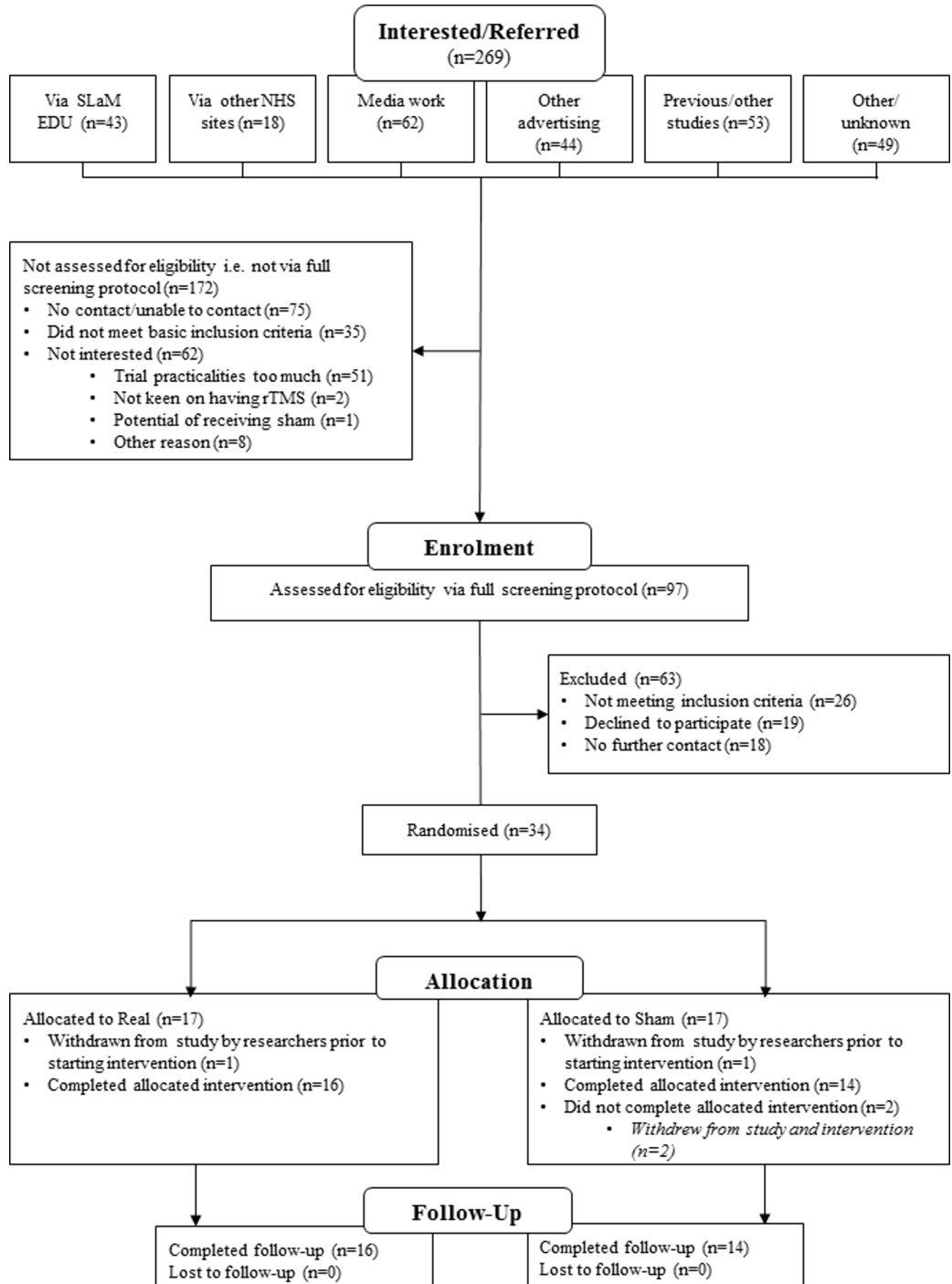

**Figure 1** CONSORT diagram of participant involvement. CONSORT, Consolidated Standards of Reporting Trials; SLaM EDU, South London and Maudsley NHS Foundation Trust Eating Disorders Unit; NHS, National Health Service; rTMS, repetitive transcranial magnetic stimulation.

other participants (n=30) completed treatment (defined a priori as ≥17 sessions of rTMS, n=18 attended the full 20 sessions) and all three study assessments, which gave a retention rate of 93.75% (30/32).

Baseline demographics and clinical characteristics are presented in table 1. All participants were female and had a long-standing illness, having previously spent a mean of nearly 11 months as an inpatient for their ED.

### Treatment effect sizes

The means, standard deviations and between-group treatment effect sizes (with confidence intervals) for change scores (baseline to post-treatment and baseline to follow-up) of the secondary clinical outcomes are presented in table 2. Group differences in BMI and ED symptoms were of small effect at both post-treatment and follow-up but favoured active treatment. At 4 months

**Table 1** Baseline demographics and clinical characteristics

| | Whole sample | | Real rTMS | | Sham rTMS | |
|---|---|---|---|---|---|---|
| | N | | N | | N | |
| Demographic details | | | | | | |
| Age (mean [SD]) | 34 | 29.74 (10.35) | 17 | 28.47 (9.48) | 17 | 31.00 (11.29) |
| Highest level of education achieved | 33 | | 17 | | 16 | |
| GCSE | 3 | | 2 | | 1 | |
| AS levels and above | 30 | | 15 | | 15 | |
| Ethnicity | 34 | | 17 | | 17 | |
| White | 31 | | 16 | | 15 | |
| Other | 3 | | 1 | | 2 | |
| Marital status | 34 | | 17 | | 17 | |
| Single | 26 | | 13 | | 13 | |
| Married | 6 | | 4 | | 2 | |
| Divorced | 1 | | 0 | | 1 | |
| Other | 1 | | 0 | | 1 | |
| Clinical characteristics | | | | | | |
| Diagnosis | 34 | | 17 | | 17 | |
| AN-R | 22 | | 11 | | 11 | |
| AN-BP | 12 | | 6 | | 6 | |
| BMI, kg/m$^2$ (mean [SD]) | 33 | 16.00 (1.44) | 17 | 15.76 (1.62) | 16 | 16.26 (1.22) |
| Duration of illness, years (mean [SD]) | 34 | 14.07 (10.75) | 17 | 13.74 (10.74) | 17 | 14.41 (11.09) |
| Number of previous ED hospitalisations (mean [SD]) | 34 | 2.18 (1.91) | 17 | 2.47 (2.07) | 17 | 1.88 (1.76) |
| Number of previous ED inpatient stays, months (mean [SD]) | 33 | 10.49 (11.66) | 17 | 12.37 (12.46) | 16 | 8.50 (10.78) |
| Previous course (≥1) of ED outpatient treatment | 29 | | 14 | | 15 | |
| Previous course (≥1) of ED day-patient treatment | 19 | | 10 | | 9 | |
| Current treatment | 34 | | 17 | | 17 | |
| Receiving ED day-patient treatment | 2 | | 1 | | 1 | |
| Receiving ED outpatient treatment | 25 | | 13 | | 12 | |
| Receiving no treatment | 7 | | 3 | | 4 | |
| Antidepressant medication | 21 | | 11 | | 10 | |
| Other psychotropic medication | 7 | | 2 | | 5 | |
| Antipsychotic medication | 4 | | 1 | | 3 | |
| Benzodiazepine/other anxiolytic/ sedative medication | 4 | | 2 | | 2 | |
| EDE-Q Global (mean [SD]) | 33 | 4.16 (1.11) | 17 | 4.07 (1.28) | 16 | 4.25 (0.94) |
| EDE-Q Restraint Subscale (mean [SD]) | 33 | 3.99 (1.54) | 17 | 3.87 (1.46) | 16 | 4.11 (1.65) |
| EDE-Q Eating Concern Subscale (mean [SD]) | 33 | 3.65 (1.21) | 17 | 3.59 (1.45) | 16 | 3.71 (0.93) |
| EDE-Q Shape Concern Subscale (mean [SD]) | 33 | 4.68 (1.31) | 17 | 4.58 (1.55) | 16 | 4.78 (1.03) |
| EDE-Q Weight Concern Subscale (mean [SD]) | 33 | 4.33 (1.25) | 17 | 4.28 (1.33) | 16 | 4.38 (1.20) |
| CIA Total (mean [SD]) | 33 | 43.64 (11.36) | 17 | 43.35 (12.72) | 16 | 43.94 (10.12) |

Continued

Table 1 Continued

| | Whole sample | | Real rTMS | | Sham rTMS | |
|---|---|---|---|---|---|---|
| | N | | N | | N | |
| EQ-5D-5L: how good or bad is your health today? (mean [SD]) | 33 | 48.91 (17.44) | 17 | 47.47 (18.63) | 16 | 50.44 (16.55) |
| DASS-21 Depression (mean [SD]) | 33 | 26.12 (9.68) | 17 | 26.82 (9.44) | 16 | 25.38 (10.19) |
| DASS-21 Anxiety (mean [SD]) | 33 | 15.39 (10.29) | 17 | 14.82 (8.31) | 16 | 16.00 (12.31) |
| DASS-21 Stress (mean [SD]) | 33 | 26.91 (7.92) | 17 | 28.35 (7.12) | 16 | 25.38 (8.66) |
| DASS-21 Total (mean [SD]) | 33 | 68.42 (24.52) | 17 | 70.00 (20.59) | 16 | 66.75 (28.72) |
| POMS Total Mood Disturbance (mean [SD]) | 33 | 83.97 (36.75) | 17 | 81.41 (36.84) | 16 | 86.69 (37.66) |
| OCI-R total (mean [SD]) | 33 | 27.79 (16.97) | 17 | 24.00 (16.48) | 16 | 31.81 (17.05) |

SD, standard deviation; GCSE, General Certificate of Secondary Education; AS, Advanced Subsidiary; AN-BP, anorexia nervosa binge–purge subtype; AN-R, anorexia nervosa restrictive subtype; CA, California; CIA, Clinical Impairment Assessment; DASS-21, Depression, Anxiety and Stress Scale – 21 items; ED, eating disorder; EDE-Q, Eating Disorder Examination Questionnaire; EQ-5D-5L, EuroQol Quality of Life Scale; POMS, Profile of Mood States; OCI-R, revised Obsessive-Compulsive Inventory; rTMS, repetitive transcranial magnetic stimulation.

post-randomisation, there were between-group differences of medium to large effect size in measures of mood, obsessive compulsive symptoms and quality of life, all favouring the active treatment. The adjusted means for the planned future primary outcome, DASS total score, were −21.25 (SD 24.33) in the real intervention group and −3.75 (SD 12.75) in the sham group, with a between-group effect size of $d=-0.9$ (95% CI −1.62 to −0.17).

At baseline, 25 participants received outpatient treatment and 5 were not receiving treatment. One participant per group received day-patient treatment. A high proportion of participants were taking antidepressants and remained on this at a stable dose throughout the trial. Participants' utilisation of TAU is shown in table 1.

At follow-up, the two participants originally in day-patient treatment were instead receiving outpatient treatment. Three participants had increased treatment intensity at follow-up, with two (one per group) starting inpatient treatment and one (from the real group) starting day-patient treatment. Of the remaining participants, two initially receiving no treatment started outpatient treatment and eight decreased intensity from outpatient treatment to no treatment.

Of those who completed the sham intervention, 71% took up the offer of having real rTMS treatment.

### Safety

In addition to the one withdrawn participant whose weight dropped below range prior to starting treatment, one other participant's weight (from the real group) was recorded below BMI $14 \, kg/m^2$ ($13.80 \, kg/m^2$) in their final rTMS session. No other participants' weight fell below the required BMI range for the duration of treatment. Blood pressure and pulse measurements did not raise any undue concerns during the study. One participant had lowered baseline potassium and start of treatment was delayed by 1 week. Blood samples for the remaining participants raised no major concerns, that is,

termination or postponing of treatment was not required. For side effects experienced, see online supplementary table 1.

### DISCUSSION

#### Principal findings

The main findings relate to the primary feasibility objectives of this study. We were able to recruit participants as planned, after making an adjustment to recruitment numbers. Many people interested in the trial could not be recruited as travelling to London for rTMS sessions proved impractical. A future trial therefore needs to consider offering treatment in several centres with easy transport access. Research follow-up rates exceeded our prespecified criterion of ≥80%. Treatment session attendance was excellent in both groups. Although for pragmatic reasons, and compared with others, our definition of 'severe, enduring AN' was lenient,[19] we managed to recruit and retain a very chronic and treatment-refractory population.

In relation to our secondary feasibility objectives, there were large between-group effect sizes on change scores from pre-treatment to follow-up on several mood variables (eg, DASS global score $d=-0.9$, 95% CI −1.62 to −0.17), favouring real rTMS. Comorbid depression is common in AN and has been shown to be associated with poor quality of life in people with SE-AN.[4] The importance of improving quality of life in SE-AN, rather than focusing on changing ED symptoms and weight gain has been emphasised,[36] and the improvements in depression observed here may contribute to the broader aim of enhancing quality of life in this group. Also, given that antidepressants are typically not very effective in underweight populations or have unacceptable side effects,[1] rTMS may provide an alternative treatment for common comorbid symptoms such as depression and anxiety. Within the current study, a higher proportion of participants were

**Table 2** The mean change scores (post-treatment and follow-up scores adjusted for baseline) for the secondary clinical outcome measures, including the number of participants included in the analysis (N), means and SD for each group and the estimated effect size (Cohen's *d*) with 95% CIs

| Assessment | Post-treatment (adjusted for baseline) | | | | | | | Follow-up (adjusted for baseline) | | | | | | |
| | Real | | | Sham | | | | Real | | | Sham | | | |
| | N | Mean | SD | N | Mean | SD | *d* (95% CI) | N | Mean | SD | N | Mean | SD | *d* (95% CI) |
|---|---|---|---|---|---|---|---|---|---|---|---|---|---|---|
| **ED-related outcomes** | | | | | | | | | | | | | | |
| BMI | 16 | 0.11 | 0.73 | 16 | −0.08 | 0.32 | 0.33 (−0.37 to 1.03) | 16 | 0.28 | 1.25 | 16 | 0.04 | 1.05 | 0.20 (−0.49 to 0.90) |
| EDE-Q Global | 16 | −0.28 | 0.73 | 16 | −0.40 | 0.79 | 0.16 (−0.54 to 0.85) | 16 | −0.43 | 0.83 | 16 | −0.52 | 0.87 | 0.10 (−0.60 to 0.79) |
| Self-Starvation Scale | 16 | −4.00 | 16.15 | 16 | −6.81 | 13.53 | 0.19 (−0.51 to 0.88) | 16 | −13.06 | 20.78 | 16 | −9.60 | 13.68 | −0.20 (−0.89 to 0.50) |
| FoFM Anxiety About Eating | 16 | −2.94 | 6.35 | 16 | −4.44 | 8.97 | 0.19 (−0.50 to 0.89) | 16 | −4.69 | 6.39 | 16 | −4.56 | 8.74 | −0.02 (−0.71 to 0.68) |
| FoFM Food Avoidance Behaviours | 16 | −3.56 | 5.77 | 16 | −3.00 | 3.39 | −0.12 (−0.81 to 0.58) | 16 | −3.50 | 6.00 | 16 | −1.69 | 5.51 | −0.32 (−1.01 to 0.39) |
| FoFM Feared Concerns | 16 | −2.00 | 6.50 | 16 | −1.81 | 9.74 | −0.02 (−0.72 to 0.67) | 16 | −2.63 | 6.79 | 16 | −1.50 | 9.25 | −0.14 (−0.83 to 0.56) |
| EDRSQ Normative Eating Self-Efficacy | 16 | 0 | 0.56 | 16 | 0.13 | 0.62 | −0.22 (−0.92 to 0.47) | 16 | 0.26 | 0.77 | 16 | 0.29 | 0.59 | −0.04 (−0.73 to 0.66) |
| EDRSQ Body Image Self-Efficacy | 16 | −0.08 | 0.49 | 16 | −0.11 | 0.61 | 0.06 (−0.63 to 0.75) | 16 | 0.08 | 0.47 | 16 | 0.16 | 0.61 | −0.14 (−0.83 to 0.56) |
| **Clinical impairments/quality of life** | | | | | | | | | | | | | | |
| CIA | 16 | −6.31 | 12.37 | 16 | −4.69 | 5.87 | −0.17 (−0.86 to 0.53) | 16 | −9.56 | 15.66 | 16 | −6.00 | 9.19 | −0.28 (−0.97 to 0.42) |
| EQ-5D-5L: how good or bad is your health today? | 8 | −0.25 | 19.65 | 10 | 7.70 | 16.67 | −0.44 (−1.38 to 0.51) | 16 | 13.06 | 18.31 | 16 | 4.81 | 13.15 | 0.52 (−0.19 to 1.22) |
| **Mood/affect/anxiety** | | | | | | | | | | | | | | |
| DASS-21 Depression | 16 | −5.13 | 8.94 | 16 | −3.25 | 10.55 | −0.19 (−0.89 to 0.51) | 16 | −9.13 | 10.61 | 16 | −1.13 | 8.58 | **−0.83 (−1.55 to −0.10)** |
| DASS-21 Anxiety | 16 | −7.25 | 6.15 | 16 | −4.13 | 5.44 | −0.54 (−1.24 to 0.17) | 16 | −4.88 | 7.19 | 16 | −1.00 | 4.79 | −0.63 (−1.34 to 0.08) |
| DASS-21 Stress | 16 | −6.75 | 9.26 | 16 | −4.50 | 4.82 | −0.31 (−1.00 to 0.40) | 16 | −7.25 | 9.71 | 16 | −1.63 | 3.88 | **−0.76 (−1.47 to −0.04)** |
| DASS-21 Total | 16 | −19.13 | 21.80 | 16 | −11.88 | 17.73 | −0.37 (−1.06 to 0.34) | 16 | −21.25 | 24.33 | 16 | −3.75 | 12.75 | **−0.90 (−1.62 to −0.17)** |
| PANAS Positive Affect | 16 | 1.75 | 5.23 | 16 | 1.06 | 5.40 | 0.13 (−0.57 to 0.82) | 16 | 4.56 | 5.79 | 16 | 0.13 | 3.88 | 0.90 (0.17 to 1.62) |
| PANAS Negative Affect | 16 | −3.81 | 9.40 | 16 | −1.44 | 5.63 | −0.31 (−1.00 to 0.39) | 16 | −7.00 | 9.13 | 16 | −1.94 | 7.42 | −0.61 (−1.31 to 0.11) |
| POMS total mood disturbance | 16 | −9.88 | 37.68 | 16 | −8.06 | 21.20 | −0.06 (−0.75 to 0.63) | 16 | −36.75 | 39.08 | 16 | −5.50 | 20.82 | **−1.00 (−1.73 to −0.25)** |
| OCI-R total | 16 | −3.69 | 7.55 | 16 | 0.94 | 5.58 | −0.70 (−1.41 to 0.02) | 16 | −1.88 | 8.13 | 16 | 0.81 | 7.84 | −0.34 (−1.03 to 0.36) |

Bold font signifies that the CI do not include 0.
BMI, bodymass index; CIA, Clinical Impairment Assessment; *d*, Cohen's *d*; DASS-21, Depression, Anxiety and Stress Scale – 21 items; ED, eating disorder; EDE-Q, Eating Disorder Examination Questionnaire; EDRSQ, Eating Disorder Recovery Self-Efficacy Questionnaire; EQ-5D-5L, EuroQol Quality of Life Scale; FoFM, Fear of Food Measure; OCI-R, revised Obsessive-Compulsive Inventory; PANAS, Positive and Negative Affect Schedule; POMS, Profile of Mood States.

taking antidepressant medication (61.7%; 21/34 participants) and somewhat higher depression scores were observed, compared with other treatment studies of AN.[37 38] This may suggest that either our participants had particularly high levels of comorbid depression or that we attracted participants who were particularly drawn to 'physical/biologically targeted treatments' rather than psychological treatments. Having said that, many participants had previously undertaken unsuccessful psychological treatments.

We considered that rTMS may be interacting with the actions of the medication to produce this antidepressant effect; however, there is no evidence for this mechanism in the depression literature. Developing better evidence for the treatment of comorbidities in EDs is a research recommendation in the recent NICE guidelines[18] and, therefore, our study potentially fills an important gap.

In addition to the mood effects, there were medium between-group effect sizes on follow-up change scores in quality of life ($d=0.52$, 95% CI −0.19 to 1.22), whereas between-group effect sizes on change scores for BMI ($d=0.2$, 95% CI −0.49 to 0.90) and ED symptoms ($d=0.1$, 95% CI −0.60 to 0.79) were small. Larger between-group effect sizes were seen on change scores from pre-treatment to follow-up than to post-treatment, suggesting that changes develop over time, rather than being due to immediate effects of rTMS. A similar delay in effect was observed in our previous case series of rTMS in SE-AN.[11] rTMS was safe, well tolerated and considered to be an acceptable treatment by participants. These various findings suggest that it is feasible to conduct a future larger scale therapeutic RCT with a sham-controlled design to establish the therapeutic efficacy of rTMS in SE-AN.

## Strengths and limitations

Our study has several strengths. It is the first RCT of multi-session rTMS treatment in individuals with AN. Second, it focused on people with severe, enduring illness. As such, it adds to the limited number of studies that have specifically targeted people with SE-AN. Third, it was sham-controlled, which is considered the gold standard method of evaluating the clinical efficacy of rTMS treatment.[39] Fourth, the majority of participants did not correctly guess their treatment allocation at follow-up, suggesting blinding was successful (see online supplementary figures 1 and 2). Lastly, the rTMS was individualised through the use of neuronavigation and a wide range of measures to assess relevant clinical outcomes were used.

In terms of limitations, the duration of the follow-up period was relatively short.[11] Our choices regarding the rTMS protocol and target brain area (left DLPFC) were theoretically, evidentially and practically based[12]; however, the optimal brain areas to target and the rTMS protocols to administer in SE-AN are unknown. We used a shorter illness duration (minimum of 3 years) than what is commonly used to define SE-AN (eg, 7 years[19]),

but nonetheless managed to recruit participants with a long-standing illness who had typically received several previous courses of intensive treatment. Our attempts to keep researchers blind to treatment allocation were only partly successful; approximately 20% of rTMS sessions were delivered by an unblinded researcher, and another researcher correctly guessed treatment allocation of participants.

## Strengths and limitations in relation to other studies

Research into treatments for people with SE-AN is limited.[36] In addition to this study, there have only been two trials with a focus on SE-AN. The first of these assessed the efficacy of 30 sessions of cognitive behavioural therapy for AN compared with specialist supportive clinical management in 63 patients.[6] Between-group differences in clinical outcomes were minimal. Within-group assessments showed small to moderate effect sizes for BMI change and medium to large for ED symptoms, depression and quality of life from baseline to end of treatment and to 6 month and 12 month follow-up. The second study investigated the effects of 4 weeks of a synthetic cannabinoid agonist (dronabinol) versus placebo as an adjunct to a multimodal treatment combining psychotherapy with nutritional interventions in 25 patients with SE-AN.[7] Dronabinol produced significantly greater short-term weight gain than placebo, but changes in ED symptoms were minimal during the study period. No follow-up data were reported. In both of these studies, treatment drop-out rates were low, as in the current study, highlighting the desire of people with SE-AN to participate in novel treatments.

## Implications for future research

Building on the present study, a large-scale multicentre RCT of real versus sham rTMS as an adjunct to TAU with a similar design should be considered. Such a trial should include a longer follow-up period (eg, 6 months and 12 months) to assess the persistence or otherwise of rTMS effects. This is also of importance given that neuroplastic changes develop over time.[40] For example, studies of deep brain stimulation in AN have shown that in treatment responders, changes in mood predate those in ED symptoms by several months (eg, [41]). Relatedly, it would be desirable to include multimodal assessment of comorbidities, for example, using a combination of semi-structured interviews, observed-rated measures and self-reports. Second, an assessment of the cost-effectiveness of establishing rTMS as a treatment option for SE-AN should be carried out. Inclusion of inpatients with SE-AN in a future trial would be desirable, as it would be easier for them to attend daily sessions. This might also allow inclusion of patients with a BMI <14 kg/m$^2$, given that inpatients have regular medical monitoring and that their food intake is more regular than that of community-dwelling patients.

Several questions need to be considered in future research of rTMS in SE-AN. The optimal brain areas to target and the rTMS protocols for SE-AN are not known.

High-frequency rTMS targeting the DLPFC was chosen for the current study as it was hypothesised that this would remediate the hypoactivity observed in AN in response to symptom provocation, cognitive flexibility and set-shifting tasks, and thus rebalance cognitive control and reward systems.[12] It was also selected given the strong evidence base for high-frequency DLPFC rTMS in other neuro-circuit-based disorders (eg, treatment-resistant depression[42]). Following on from research in depression, the use of low-frequency rTMS or intermittent theta burst stimulation (iTBS) in comparison with high-frequency rTMS might be tested. Low-frequency rTMS is thought to have fewer side effects and be more well tolerated, and iTBS would substantially reduce treatment time and participant burden. In the depression literature, it appear that both of these have similar levels of efficacy to high-frequency rTMS.[43–46] Future studies should also consider rTMS as an adjunct to psychological therapies.[47] Other neuromodulation treatments in combination with cognitive interventions have shown promise,[48 49] and so, addition of rTMS to structured psychotherapy or cognitive training tasks in SE-AN may help increase its efficacy.[12] Finally, additional work on neural and neurocognitive mechanisms of action of rTMS and the cost-effectiveness of this treatment are necessary.

## CONCLUSION

In this feasibility RCT, rTMS was safe and well tolerated. This study provides preliminary evidence for the therapeutic potential of rTMS treatment in community-dwelling SE-AN as an adjunct to TAU. It suggests that it is feasible to conduct a future larger scale therapeutic RCT with a sham-controlled design to establish/confirm the therapeutic efficacy of rTMS in AN. The findings from this trial will inform a future large-scale RCT with respect to decisions on primary outcome measures and other aspects of protocol development, such as sample size, design, location and number of research centres. Future studies should include a longer follow-up period and a formal assessment of cost-effectiveness. Consideration should also be given to use of alternative stimulation protocols (eg, low-frequency rTMS) and the combination of rTMS and ED-specific therapies/tasks to maximise impact on ED and mood.

**Author affiliations**
[1]Section of Eating Disorders, Department of Psychological Medicine, Institute of Psychiatry, Psychology & Neuroscience, King's College London, London, UK
[2]Department of Neuroimaging, Centre for Neuroimaging Sciences, Institute of Psychiatry, Psychology & Neuroscience, King's College London, London, UK
[3]Department of Biostatistics and Health Informatics, Institute of Psychiatry, Psychology & Neuroscience, King's College London, London, UK
[4]Department of Psychosis Studies, Institute of Psychiatry, Psychology & Neuroscience, King's College London, London, UK
[5]Eating Disorders Service, South London and Maudsley NHS Foundation Trust, London, UK

**Acknowledgements** We would like to thank the individuals who participated in this study for their time and commitment and our patient and public involvement advisors for their valuable feedback. The authors would also like to thank Daniela Mercado Beivide and Pia Lauffer for their assistance in data collection, the radiographers at the KCL Centre for Neuroimaging Sciences for their work during with the MRI scans and their guidance on decisions regarding eligibility and protocol development and the phlebotomy staff at South London and Maudsley Outpatients. This study was supported by the United Kingdom Clinical Research Collaboration-registered King's Clinical Trials Unit at King's Health Partners, which is part-funded by the National Institute for Health Research Biomedical Research Centre for Mental Health at South London and Maudsley NHS Foundation Trust and King's College London and the NIHR Evaluation, Trials and Studies Coordinating Centre.

**Contributors** BD, SB, JM and US drafted the manuscript. BD and SB conducted data analysis, which was supervised by BC and US. OGO'D, ICC, MK, SJR, NK, DG and ASD revised the manuscript critically for important intellectual content. Ethical approval was obtained by SB, JM, MK and US. Funding from National Institute for Health Research (NIHR) was obtained by US, JM and ICC. Funding from the NIHR Biomedical Research Centre (BRC) was obtained by SB, JM, MK, OGO'D, ICC and US. JM, SB and US registered the trial on the International Standard Randomised Controlled Trial Number (ISRCTN) registry. JM, SB, MK, US and BD were involved in participant recruitment. BD, SB, JM, MK, JW and SJR were involved in data collection. rTMS treatment was provided by BD, SB, JM and MK. OGO'D, MK, JW, SJR, SB, JM, ICC and US contributed to the design and conception of the study. All authors were involved in drafting, critiquing and approving of the manuscript and accept responsibility for the accuracy and integrity of this work.

**Funding** This work was supported by a National Institute for Health Research (NIHR) Research for Patient Benefit (RfPB) grant (RB-PG-1013-32049) and Infrastructure Support for Pilot studies from the NIHR Biomedical Research Centre at South London and Maudsley NHS Foundation Trust (SLaM) and King's College London. This paper presents independent research funded by the NIHR under its RfPB Programme (Grant Reference Number PB-PG-1013-32049).

**Disclaimer** The views expressed are those of the author(s) and not necessarily those of the NHS, the NIHR or the Department of Health.

**Competing interests** OGO receives salary support from an NIHR Infrastructure grant for the Wellcome Trust/KCL Clinical Research Facility. Ulrike Schmidt is supported by an NIHR Senior Investigator Award. US, ASD and ICC receive salary support from the NIHR Mental Health BRC at SLaM NHS Foundation Trust and KCL.

**Patient consent** Obtained.

**Ethics approval** Ethical approval was given by the London – City Road & Hampstead Research Ethics Committee (ref: 15/LO/0196).

**Provenance and peer review** Not commissioned; externally peer reviewed.

**Data sharing statement** No additional data are available.

**Author note** The study protocol was published prior to recruitment.

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
