## [Reviewer comments · BMJ Open]

ARTICLE DETAILS

TITLE (PROVISIONAL)	A randomised controlled feasibility trial of real versus sham repetitive transcranial magnetic stimulation treatment in adults with severe and enduring anorexia nervosa: the TIARA study.
AUTHORS	Dalton, Bethan; Bartholdy, Savani; McClelland, Jessica; Kekic, Maria; Rennalls, Samantha; Werthmann, Jessica; Carter, B; O'Daly, Owen; Campbell, Iain; David, Anthony; Glennon, Danielle; Kern, Nikola; Schmidt, Ulrike

VERSION 1 – REVIEW

REVIEWER	Rebecca J Park Oxford University , UK
REVIEW RETURNED	06-Mar-2018

GENERAL COMMENTS	This manuscript describes some results from the first feasibility RCT of TMS in patients who have had Anorexia Nervosa for longer than 3 years. It also documents clinical outcome at 4 months. Consistent with the documented effects of TMS on depression, which is highly comorbid in AN, these findings suggest that TMS in AN improves mood, as it does for depressed people without AN . This is unsurprising but worth documenting, especially given antidepressants are not always effective in AN. That said, it is unclear how TMS will affect the AN psychopathology over a longer time period. The paper is well written and in general well executed, but would be optimised by suggested revisions outlined below:  • rTMS has been most widely and successfully used for and found effective for major depression . Did the authors have any specific hypotheses or expectations about why TMS would benefit AN pathology specifically, or is TMS expected to be more an alternative approach to improve mood in this hard to treat group? • A major limitation is the poor documentation and assessment of comorbidities. Details of comorbid diagnoses appear to be lacking e.g. MDD, PTSD, OCD, and past treatment thereof. It would have been optimal to assess ED and comorbid pathologies using observer rated and semi structured interviews such as SCID and EDE. This is especially important in AN as self-reports are often unreliable. Given the research evidence that rTMS improves depression, a table of comorbid psychiatric diagnoses, and in particular the numbers of participants in true vs sham groups with current or past major depressive disorder, would be helpful. • Comprehensive assessment of comorbidities for future trials should ideally be multimodal, including for example of semi
---

structured interviews such as the EDE and SCID, and observer rated measures such as Hamilton Depression and anxiety scales HAM A D, to augment self-reports.

- The term severe enduring AN is typically used to describe individuals who have been ill longer than 7 years. The authors may wish to consider a different label for the AN group in this paper, who have been ill longer than 3 years. It is unclear how severe their AN was given some have BMIs 16-18 and no EDE subscales are reported.

- Given neurogenesis can take time, the follow-up period of 4 months is likely inadequate to fully determine rTMS on AN pathology as such changes may take longer than this. Consider for example published studies of Deep brain stimulation for AN to mood regulatory targets, which suggest that in those that respond, mood and anxiety change first followed only later over many months by ED changes. Eg see Lipsman : Lancet Psychiatry 2017; 4: 285–94

- It is unfortunate the actual N of 32 rather than desired 44 was achieved. Given the large interest in the study (269 potential participants screened) the authors may wish to expand further on how the group recruited may differ from those non recruited, and how this reflects the pragmatics of recruitment and delivery of r TMS as a potential intervention for AN

- Line 39: It is unfortunate on researcher unblinded patients. Could analyses be confirmed excluding these unblinded sessions?

- How confident are the authors that any DASS effects are real and not just a reflection of inadequate other forms of support for mood and stress?

- Future studies would benefit from more homogenous study groups, for example focussing on either restrictive or binge pure subtypes of AN, and/ or those with or without comorbid MDD diagnoses. [Given suggestions from prior literature of TMS effects on depression, and on binge eating in particular treatment effects are more likely in this subtype especially if depressed.

- I note the sham group have on average a longer illness and lower BMI, yet fewer past treatment months. Please comment on this in limitations. Even these small differences in duration and weight and treatment could reflect on responsiveness, and or mean this group are actually more depressed.

- In clinical practice the lower BMI goes the more treatment intractable is both AN and accompanying depression. It would be interesting to address the question of whether TMS can augment existing therapies in this lower BMI group, who are more likely to be unresponsive to antidepressants and psychological treatments. Subsidiary analyses could usefully look at subgroups with BMI > < 15, or compare those above or below median BMI .

- A mention of other forms of neuromodulation in AN , eg DBS ,

	TDCS, could be added, and the pros and cons of non-invasive strategies.  • Given it is all too easy to inadvertently exploit the hope of patients who have exhausted other treatment options, the authors may want to consider augmenting future trials with a robust neuro-ethical framework, as suggested for invasive forms of neuromodulation (Park, Tan, Singh 2017). While medical risks are less for TMS there are otherwise similarities in the issues to be considered in brain based treatments. • Supplementary files: Typo in 3rd line of blinding results 'think' meant to read 'thick'?
--	--

REVIEWER	S Guillaume chu and university of Montpellier France
REVIEW RETURNED	09-Mar-2018

GENERAL COMMENTS	This feasibility and exploratory study is the first RCT of multi-session rTMS treatment in individuals with AN. It was sham-controlled with successful blinding, which is considered the gold-standard method of evaluating the clinical efficacy. It focused on people with severe, enduring illness for whom we lack effective treatment. As such, it adds to the limited number of studies that have specifically targeted people with SE-AN which increases the interest of this study. In this feasibility RCT, rTMS was safe and well tolerated. This study provides preliminary evidence for the therapeutic potential of rTMS treatment in community-dwelling SE-AN as an adjunct to TAU, even if in terms of clinical outcomes, the results regarding clinical outcomes are limited. Nevertheless, well-designed and well-executed studies like this are very valuable and they can inform future approaches testing the use of rTMS with different targets or parameters. Here are some suggestions:  1) Regarding the effect size, the standard deviations of the confidence intervals are wide and asymmetrical. Was the distribution of the variables Gaussian? If not, is it possible to consider a transformation of the variables? 2) Beyond a treatment in day hospital or in outpatient, there is very little information on the usual treatment of which these patients benefited. One or two sentences describing the treatment as usual would be useful? 3) Was there some other medication beyond antidepressant (such as olanzapine)? 4) The authors rightly discuss the question of the ineffectiveness of antidepressants in patients with low BMI. Was there any interaction between mood response in active group and taking antidepressant? 5) Given the primary outcome was acceptance and feasibility. Based on the flow chart a high number of patients refused the protocol (62 + 19). This should be discussed.
---

VERSION 1 – AUTHOR RESPONSE

Reviewer: 1

□ This manuscript describes some results from the first feasibility RCT of TMS in patients who have had Anorexia Nervosa for longer than 3 years. It also documents clinical outcome at 4 months. Consistent with the documented effects of TMS on depression, which is highly comorbid in AN, these findings suggest that TMS in AN improves mood, as it does for depressed people without AN. This is unsurprising but worth documenting, especially given antidepressants are not always effective in AN. That said, it is unclear how TMS will affect the AN psychopathology over a longer time period. The paper is well written and in general well executed, but would be optimised by suggested revisions outlined below:

We thank the reviewer for this kind comment and have addressed issues on a point by point basis.

□ rTMS has been most widely and successfully used for and found effective for major depression. Did the authors have any specific hypotheses or expectations about why TMS would benefit AN pathology specifically, or is TMS expected to be more an alternative approach to improve mood in this hard to treat group?

Tentative hypotheses and rationale were thoroughly detailed in the protocol paper for the study (Bartholdy et al., 2015. *Trials*, 16:548). Due to the word limit, we did not expand on this in the current paper, however, we have now more clearly directed readers to this protocol paper in the introduction (page 3, paragraph 6): “The study rationale, aims, and tentative hypotheses, along with the trial design and methodology are described fully in a protocol paper [16]”.

□ A major limitation is the poor documentation and assessment of comorbidities. Details of comorbid diagnoses appear to be lacking e.g. MDD, PTSD, OCD, and past treatment thereof. It would have been optimal to assess ED and comorbid pathologies using observer rated and semi structured interviews such as SCID and EDE. This is especially important in AN as self-reports are often unreliable. Given the research evidence that rTMS improves depression, a table of comorbid psychiatric diagnoses, and in particular the numbers of participants in true vs sham groups with current or past major depressive disorder, would be helpful.

We thank the reviewer for highlighting this. When we planned the study we did consider the use of semi-structured interviews such as the EDE interview and SCID, but on balance decided against this for a number of reasons. Most importantly, our protocol for this study is already very demanding on participants and we did not want to add additional burden by including lengthy interviews to assess eating disorder symptoms and comorbid disorders. Whilst we are aware that EDE-Q and interview are not interchangeable, they do measure the same constructs, and for binge-purge behaviours, which is of greatest relevance here, to determine eating disorder sub-type, the agreement between questionnaire and interview is good. Furthermore, we have included BMI as an objective measure of eating disorder severity and we also used several task-based measures as objective measures which will be published elsewhere. This is addressed further in relation to the next point raised.

While we used a brief version of the SCID to screen out people with psychosis or substance dependence, we did not systematically collect information on all comorbid diagnoses. Therefore, we do not have the relevant data to report.

□ Comprehensive assessment of comorbidities for future trials should ideally be multimodal, including for example of semi structured interviews such as the EDE and SCID, and observer rated measures such as Hamilton Depression and anxiety scales HAM A D, to augment self-reports.

We agree that this would be an important consideration for future trials. To acknowledge this, we have added the following sentence into the Implications for Future Research section of the Discussion (page 13, paragraph 1): “Relatedly, it would be desirable to include multimodal assessment of comorbidities, for example, using a combination of semi-structured interviews, observed-rated measures and self-reports.”

□ The term severe enduring AN is typically used to describe individuals who have been ill longer than 7 years. The authors may wish to consider a different label for the AN group in this paper, who have been ill longer than 3 years. It is unclear how severe their AN was given some have BMIs 16-18 and no EDE subscales are reported.

We agree with the reviewer that this is an important issue. Currently, there does not seem to be a consensus on what constitutes SE-AN (Broomfield et al., 2017. *Int J Eat Disord*. 2017, 50:611-623). Most definitions emphasise a sustained period of significant underweight (body mass index (BMI) <17.5 kg/m²) from > 3 to > 10 years (Treasure & Russell, 2011. *Br J Psychiatry*, 199:5-7; Hay et al., 2012. *Aust N Z J Psychiatry*, 46:1136-44; Arkell & Robinson, 2008. *Int J Eat Disord*, 41:650-6; Touyz et al., 2013. *Psychol Med*, 43:2501-11; Tierney & Fox, 2009. *Int J Eat Disord*, 42:62-7; Long et al., 2012. *Clin Psychol Psychother*, 19:1-13; Wonderlich et al., 2012. *Int J Eat Disord*, 45:467-75). Converging evidence suggests that after a period of > 3-5 years at low weight, responsiveness to treatment and outcome of AN is poor, probably because of neurotoxic effects of starvation and stress hormones (cortisol) on the brain (Treasure & Russell, 2011. *Br J Psychiatry*, 199:5-7; Currin & Schmidt, 2005. *J Mental Health*, 14:1-14). To acknowledge the issue, we have added the following sentence into the Methods section (page 4, paragraph 2) following our definition of severe enduring anorexia nervosa: “(We accept that there is a continuing debate on definitions of SE-AN, for review see Broomfield, et al. [19])”.

Therefore, we chose a > 3 years illness duration given that this is the minimum illness duration cited in the literature after which treatment outcomes tend to be poorer. We also included an additional inclusion criterion that participants must have had at least one previous adequate course of specialist treatment (e.g. one 6-month course of specialist outpatient therapy, specialist day-care or in-patient treatment for refeeding), which further emphasises enduring AN and treatment resistance in this patient group.

We have added in the EDE-Q subscales into Table 1 to illustrate illness severity in this group.

□ Given neurogenesis can take time, the follow-up period of 4 months is likely inadequate to fully determine rTMS on AN pathology as such changes may take longer than this. Consider for example published studies of Deep brain stimulation for AN to mood regulatory targets, which suggest that in those that respond, mood and anxiety change first followed only later over many months by ED changes. Eg see Lipsman : *Lancet Psychiatry* 2017; 4: 285–94

We accept that this is important. We have commented on the short duration of the follow-up as a limitation (page 12, paragraph 1 of Strengths and Limitations section) and that future studies should include a longer follow-up period (page 13, paragraph 1 of Implications for Future Research section). We have now added in the suggested reference and added in the following sentence in the Implications for Future Research section of the discussion (page 13, paragraph 1): “This is also of importance given that neuroplastic changes develop over time [39]. For example, studies of deep brain stimulation in AN have shown that in treatment responders, changes in mood predate those in ED symptoms by several months [e.g., 40].”

□ It is unfortunate the actual N of 32 rather than desired 44 was achieved. Given the large interest in the study (269 potential participants screened) the authors may wish to expand further on

how the group recruited may differ from those non recruited, and how this reflects the pragmatics of recruitment and delivery of r TMS as a potential intervention for AN

We do not have the data to expand on how the recruited group differs from those that were not recruited. We have stated the reasons for participants not taking part in the consort diagram and we have expanded on this in the Results section by adding the following sentence (page 6, paragraph 2 of the Patient flow, attendance and retention section): “Of these, 61 (22.7%) did not meet the inclusion criteria and 81 (30.1%) were not interested and/or declined to participate, with the majority citing trial practicalities as a reason for this (e.g., accessibility, financial limitations, time commitment, etc.)” We have also already acknowledged in the Implications for future research section (page 13, paragraph 1) that future trials should be multi-centre, which will hopefully improve recruitment rates.

Line 39: It is unfortunate on researcher unblinded patients. Could analyses be confirmed excluding these unblinded sessions?

We agree that this is unfortunate, however, we were limited by practicalities within the research team. We are unable to run these analyses excluding unblinded sessions as the measures collected are at main assessment points and not by-session.

How confident are the authors that any DASS effects are real and not just a reflection of inadequate other forms of support for mood and stress?

Our study used multiple mood measures (e.g. DASS, PANAS, POMS) and they agree that there is a reduction in negative mood but also a parallel increase in positive mood. In addition, many of the participants were receiving support for their mood and/or stress (e.g. anti-depressants, psychological therapy). Furthermore, our preliminary analysis of our qualitative data (not reported here) is consistent with the DASS being a valid measure of mood.

Future studies would benefit from more homogenous study groups, for example focussing on either restrictive or binge pure subtypes of AN, and/ or those with or without comorbid MDD diagnoses. Given suggestions from prior literature of TMS effects on depression, and on binge eating in particular treatment effects are more likely in this subtype especially if depressed.

We agree that this may be an important consideration for future trials, however, as this was a feasibility trial we did not implement such strict eligibility criteria. We used stratified randomisation based on anorexia nervosa subtype and current treatment intensity, therefore, both of our treatment groups were balanced in terms of anorexia nervosa subtype (n=11 with anorexia nervosa restrictive subtype and n=6 with anorexia binge-purge subtype) and treatment intensity.

I note the sham group have on average a longer illness and lower BMI, yet fewer past treatment months. Please comment on this in limitations. Even these small differences in duration and weight and treatment could reflect on responsiveness, and or mean this group are actually more depressed.

We have checked the data in Table 1. The sham group has a slightly higher BMI (16.26 kg/m²) than the real group (15.76 kg/m²). Furthermore, the between-group differences in BMI (less than 1 BMI point) and illness duration (approx. 6 months) are negligible. It is also important to consider these results in light of the fact that this is a feasibility study with a small sample size.

With regards to previous inpatient treatment months, it is a crude measure of treatment received: many participants struggled to quantify the number of months as an inpatient and therefore this may

not be the most reliable measure. However, given that the reviewer has raised this, we have added into the table the proportion of participants who had reported receiving at least one course of NICE-recommended outpatient psychological therapy of appropriate length for the eating disorder and proportion of participants who have previously attended day patient treatment for the eating disorder to provide a fuller picture on past treatment.

□ In clinical practice the lower BMI goes the more treatment intractable is both AN and accompanying depression. It would be interesting to address the question of whether TMS can augment existing therapies in this lower BMI group, who are more likely to be unresponsive to antidepressants and psychological treatments. Subsidiary analyses could usefully look at subgroups with BMI > < 15, or compare those above or below median BMI.

We are unable to conduct subsidiary analyses for several reasons. Firstly, these outcomes are not in line with the guidance for conducting feasibility studies (Eldridge et al., 2016. *BMJ*, 355:i5339), secondly, the sample size is too small and finally, these analyses are not reported (i.e. planned for) in our protocol (Bartholdy et al., 2015. *Trials*, 16:548). However, we hope to look at this by combining the data from the real treatment group with the data of those who received real treatment following the sham treatment, giving us a larger sample size and thus allowing us to conduct these interesting subsidiary analyses.

□ A mention of other forms of neuromodulation in AN, eg DBS , TDCS, could be added, and the pros and cons of non-invasive strategies.

We think this is out of the scope of this paper and due to the word limit we are unable to include a discussion of these techniques. However, we have referred to our recent review (Dalton et al., 2017. *Current Opinion in Psychiatry*, 30:458-73) in the Introduction, in which we have discussed other forms of neuromodulation and their evidence in eating disorders.

□ Given it is all too easy to inadvertently exploit the hope of patients who have exhausted other treatment options, the authors may want to consider augmenting future trials with a robust neuro-ethical framework, as suggested for invasive forms of neuromodulation (Park, Tan, Singh 2017). While medical risks are less for TMS there are otherwise similarities in the issues to be considered in brain based treatments.

We are aware of the reviewer's paper making the case for and providing a gold standard neuroethics framework for deep brain stimulation and other neurosurgical interventions in anorexia nervosa. However, rTMS is a widely used non-invasive brain stimulation (NIBS) method and has a very different safety profile to DBS. It is also much easier to conduct randomised controlled trials of rTMS than of DBS and rTMS is widely used in other physically frail and vulnerable populations (e.g. treatment refractory depression, stroke). Thus, there is arguably less of a case for a special neuroethics framework for rTMS in anorexia nervosa, over and above ethical standards that apply to the conduct of any clinical trial (Cabrera et al., 2014. *Brain Topogr*, 27:33-45). In a separate qualitative study, we will explore some of the rTMS-related ethical issues such as authenticity, autonomy and the effect of rTMS on sense of self in AN.

In addition, it is important that rTMS procedures and parameters comply with current safety and application guidance, which was the case in our study. We have said this on page 6, third paragraph of the procedures section as follows: "All rTMS procedures and parameters used in the current study were in accordance with the current safety and application guidelines for rTMS (Rossi et al., 2009. *Neurophysiology*, 120:2008-39)."

□ Supplementary files: Typo in 3rd line of blinding results 'think' meant to read 'thick' ?

This has now been corrected.

Reviewer: 2

□ This feasibility and exploratory study is the first RCT of multi-session rTMS treatment in individuals with AN. It was sham-controlled with successful blinding, which is considered the gold-standard method of evaluating the clinical efficacy. It focused on people with severe, enduring illness for whom we lack effective treatment. As such, it adds to the limited number of studies that have specifically targeted people with SE-AN which increases the interest of this study. In this feasibility RCT, rTMS was safe and well tolerated. This study provides preliminary evidence for the therapeutic potential of rTMS treatment in community-dwelling SE-AN as an adjunct to TAU, even if in term of clinical outcomes, the results regarding clinical outcomes are limited. Nevertheless, well-designed and well-executed studies like this are very valuable and they can inform future approaches testing the use of rTMS with different targets or parameters.

We thank the reviewer for these comments.

□ Regarding the effect size, the standard deviations of the confidence intervals are wide and asymmetrical. Was the distribution of the variables Gaussian? If not, is it possible consider a transformation of the variables?

We agree that the confidence intervals are wide: this is likely due to the limited number of observations in the study. However, we checked all effect sizes and confidence intervals for asymmetry by subtracting the lower confidence interval from the effect size and also taking away the higher confidence interval from the effect size (e.g. Post-treatment BMI: $0.33 - (-0.37) = 0.7$, $0.33 - 1.03 = -0.7$) and did not find any evidence of asymmetry.

□ Beyond a treatment in day hospital or in outpatient, there are very little information on the usual treatment of which these patients benefited. One or two sentences describing the treatment as usual would be useful?

Treatment-as-usual is briefly defined in the second paragraph of the Methods section (page 4); however, due to the word limit we were not able to expand on the details. We have added into the table the proportion of participants who had reported receiving at least one course of NICE-recommended outpatient psychological therapy of appropriate length for the eating disorder and proportion of participants who have previously attended day patient treatment for the eating disorder. While this may not provide the most specific details, given that treatment history is difficult to quantify, we felt it best to summarise it in this way.

□ Was there some other medications beyond antidepressant (such as olanzapine)?

We have added current use of other psychotropic into Table 1 (Baseline demographics and clinical characteristics, page 7).

□ The authors rightly discuss the question of the ineffectiveness of antidepressants in patients with low BMI. Was there any interaction between mood response in active group and taking antidepressant?

We agree that this is an interesting thought. As this is a feasibility study with small sample sizes, we are unable to conduct analyses to consider this. However, we hope to look at this in a future paper in

which we will combine the data from the real treatment group with the data of those who received real treatment following the sham treatment, giving us a larger sample size.

□ Given the primary outcome was acceptance et feasibility. Based on the flow chart a high number of patients refused the protocol (62 + 19). This should be discussed.

As shown in the flow chart, a high number of people refused to take part, with most due to practical reasons (e.g. living far away, financial limitations, full-time work, etc.). We have expanded on this in the Results by adding the following sentence (page 6, paragraph 2 of the Patient flow, attendance and retention section): “Of these, 61 (22.7%) did not meet the inclusion criteria and 81 (30.1%) were not interested and/or declined to participate, with the majority citing trial practicalities as a reason for this (e.g., accessibility, financial limitations, time commitment, etc.).”

VERSION 2 – REVIEW

REVIEWER	Rebecca Park University of Oxford, UK
REVIEW RETURNED	17-Apr-2018
GENERAL COMMENTS	the authors have addressed adequately the comments from my prior review , within the constraints of the study design and data collected. the study has many limitations but remains worthy of publication.
REVIEWER	S Guillaume Department of Psychiatric Emergency & Acute Care, Lapeyronie Hospital, CHRU Montpellier, Montpellier, France, INSERM, Univ Montpellier, Neuropsychiatry: Epidemiological and Clinical Research, Montpellier, France.
REVIEW RETURNED	20-Apr-2018
GENERAL COMMENTS	The authors have adequately responded to comments